# Optimization of In Situ Indentation Protocol to Map the Mechanical Properties of Articular Cartilage

**DOI:** 10.3390/ma15186425

**Published:** 2022-09-16

**Authors:** Matteo Berni, Paolo Erani, Nicola Francesco Lopomo, Massimiliano Baleani

**Affiliations:** 1Laboratorio di Tecnologia Medica, IRCCS Istituto Ortopedico Rizzoli, 40136 Bologna, Italy; 2Dipartimento dell’Ingegneria dell’Informazione, University of Brescia, 25121 Brescia, Italy

**Keywords:** knee, bovine, human, cartilage, osteochondral unit, mechanical properties, mapping, indentation, in vitro, experimental procedure

## Abstract

Tissue engineering aims at developing complex composite scaffolds for articular cartilage repair. These scaffolds must exhibit a mechanical behavior similar to the whole osteochondral unit. In situ spherical indentation allows us to map the mechanical behavior of articular cartilage, avoiding removal of the underlying bone tissue. Little is known about the impact of grid spacing, indenter diameter, and induced deformation on the cartilage response to indentation. We investigated the impact of grid spacing (range: *a* to 3*a*, where *a* is the radius of the contact area between cartilage and indenter), indenter diameter (range: 1 to 8 mm), and deformation induced by indentation (constant indentation depth versus constant nominal deformation) on cartilage response. The bias induced by indentations performed in adjacent grid points was minimized with a 3*a* grid spacing. The cartilage response was indenter-dependent for diameters ranging between 1 and 6 mm with a nominal deformation of 15%. No significant differences were found using 6 mm and 8 mm indenters. Six mm and 8 mm indenters were used to map human articular cartilage with a grid spacing equal to 3*a*. Instantaneous elastic modulus E_0_ was calculated for constant indentation depth and constant nominal deformation. E_0_ value distribution did not change significantly by switching the two indenters, while dispersion decreased by 5–6% when a constant nominal deformation was applied. Such an approach was able to discriminate changes in tissue response due to doubling the indentation rate. The proposed procedure seems to reduce data dispersion and properly determine cartilage mechanical properties to be compared with those of complex composite scaffolds.

## 1. Introduction

Articular cartilage is a highly specialized connective tissue representing an integral part of the musculoskeletal system [1]. It is composed of a dense extracellular matrix–consisting of a three-dimensional collagen network, negatively charged proteoglycans and chondrocyte cells, and interstitial fluid [1]. The interaction between these constituents defines the articular cartilage mechanical response [2], accounting for its site-dependent, depth-dependent, and time-dependent behavior [3]. Addressing the joint biomechanics, the main functions of such a tissue are to ensure low friction articulation, more homogeneous contact pressure distribution, and shock absorption [4,5].

Articular cartilage behavior has been extensively investigated both by in vivo and ex vivo techniques. In vivo investigations either involve the use of arthroscopic instruments, with associated surgical risks and procedural costs, or non-invasive imaging, which offers a limited accuracy in predicting tissue mechanical features. Therefore, although in vivo investigations are more relevant from a clinical point of view, as they allow a monitoring of disease progression and treatment response, ex vivo assessments are still the most reliable and commonly used technique to determine the mechanical peculiarities of articular cartilage.

Ex vivo investigations involve testing of either the whole osteochondral unit or articular cartilage excised samples. Although tensile, shear, sliding, and torsional loading conditions have been applied, most of the experimental approaches involve compression testing [6]. The functional assessment of articular cartilage by confined or unconfined compression requires specimens with strict geometrical features, mainly related to specimen planarity and homogeneity. Therefore, tissue excision from the underlying subchondral bone is needed. Although compressive approaches provide valuable information on articular cartilage response, specific bias in the investigated behavior might be introduced by interrupting articular cartilage continuity and removing subchondral bone tissue, i.e., altering the boundary conditions peculiar to the osteochondral unit.

Ex vivo indentation is an alternative method that allows us to map the mechanical response of the articular cartilage without altering its boundary conditions [7,8]. Although both cylindrical and spherical indenters have been used, they are not equivalent. Indeed, the indenter geometry impacts strain distribution within the articular cartilage: a cylindrical indenter determines non-physiological strain concentration near the edge of the plane-ended probe, while a spherical indenter induces strain distribution closely resembling the one predicted in vivo [9,10]. Interestingly, neglecting micro- and nano-scale studies, a great variance in the indenter diameter, falling within the range 1.0 ÷ 6.35 mm, has been found in the literature [7,10,11,12,13,14,15]. Although there are at least four studies suggesting that indenter diameter with millimeter-scale size affects the response of articular cartilage [16,17,18,19], a sound assessment of the indenter diameter effect is still missing. This gap raises the question of the comparability of cartilage response to indentation across studies.

The choice of the indenter diameter also affects the minimum grid spacing allowed to map the articular cartilage response. Decreasing grid spacing is desirable as it determines an in increase in the number of map nodes, i.e., indentation points become closer to each other. Therefore, reduced grid spacing can capture abrupt and subtle changes in articular cartilage mechanics. On the other hand, too small grid spacing could introduce artefact effects due to interactions between adjacent indentations. In two studies investigating the cartilage response to indentation at a micrometer scale, the authors recommend a minimum grid spacing of 2.5*a* (82 µm) [20] and 3*a* [21], where *a* is the radius of the contact area between the articular cartilage and the spherical indenter. Such analyses have not been found for the millimeter scale. Given the hierarchical structure of the cartilage tissue, the best compromise between surface refinement and artefact reduction must be identified at the millimeter scale to properly map cartilage mechanical properties.

In addition to indenter diameter, the indentation depth determines the extension of the contact area between the indenter and the articular cartilage surface and, finally, the volume of tissue undergoing large deformation. Generally, indentation protocols reported in the literature set a constant indentation depth regardless of the thickness of the tissue, determining nominal deformation magnitude depending on articular cartilage thickness [7,11,12,13,20,22,23]. However, it has been demonstrated that cartilage response to indentation depends on deformation induced by the indenter [24]. Applying a displacement equal to a fixed percentage of the articular cartilage thickness should allow us to induce a constant nominal deformation level and determine the corresponding cartilage response. Such an approach should reduce the data dispersion. However, this approach requires the a priori measurement of articular cartilage thickness to be performed without damaging the tissue at the indentation point.

Addressing the above-mentioned open issues should allow us to acquire more consistent and comparable parameters describing the response of the articular cartilage to indentation when supported by the underlying subchondral bone tissue. The data thus achieved are essential to both describe the biomechanical behavior of the whole osteochondral unit and provide accurate information for tissue engineering aimed at developing a complex composite scaffold for articular cartilage repair. Accordingly, the primary goal of this study was to optimize the indentation procedure for assessing the mechanical response of the whole osteochondral unit, evaluating the impact of grid spacing (range: *a* to 3*a*, where *a* is the radius of the contact area between cartilage and indenter), indenter diameter (range: 1 to 8 mm), and deformation induced by indentation (constant indentation depth versus constant nominal deformation) on cartilage response.

## 2. Materials and Methods

### 2.1. Sample Collection and Management

Portions of skeletally mature bovine knees were used to develop and optimize our indentation experimental procedure. Bovine knees were collected from on-farm livestock within 24 h from culling. Parallelepiped-shaped samples—with a minimum thickness of 20 mm and a nominal area functional to the specific test—were extracted from the articular surface of ten knees (Appendix A) by using a diamond-coated blade (MDP200, Remet, Bologna, Italy). Cutting was performed in a circular saw (TR60, Remet, Bologna, Italy) under continuous water cooling to minimize tissue heating. Samples were then wrapped in gauze soaked in phosphate-buffered saline (PBS) solution 1X (7.4 pH, Life Technologies Europe B.V., Bleiswijk, The Netherlands) and frozen at −20 °C [25].

The day before testing, the frozen sample was thawed overnight at 4 °C [26,27]. The day of the test, the sample was removed from refrigerator and embedded in a 5 mm thick polymethylmethacrylate baseplate (Restray NF, S.P.D.: Mulazzano, Italy) needed to constrain it to the X-Y motorized table of the testing machine (Mach-1 V500css, Biomomentum Inc., Laval, QC, Canada). During the polymerization process, the articular cartilage surface was kept moist with a gauze soaked in PBS. The entire phase was carried out at room temperature and completed within 1 h of sample removal from the refrigerator [27]. All tests were performed maintaining tissue samples fully submerged in PBS and completed by the end of the day.

### 2.2. A Priori Estimate of the Articular Cartilage Thickness at the Indentation Point

The most straightforward method to measure articular cartilage thickness when a tissue sample is constrained to a material testing machine entails needle probing [28]. In order to obtain reliable information, we had to keep into account that needle insertion creates a discontinuity (i.e., a small hole of the order of magnitude of half a millimeter in the articular cartilage undergoing the indentation test), which may alter the articular cartilage response to indentation, especially when small-diameter indenters are used. Therefore, needle insertion on the indentation point prior to an indentation test should be avoided. Since the articular cartilage is a continuous layer, it was assumed that tissue thickness below the indentation point could be estimated by measuring the thickness in nearby positions. Articular cartilage thickness was measured in the four corners of a 6 mm × 6 mm square whose geometric center coincided with the indentation point. Therefore, the distance between the needle insertion points and the indentation point was 4.2 mm (see Section 2.3).

Articular cartilage thickness was measured by using a 27G × ½” intradermal needle attached to the actuator of the testing machine [11]. Needle insertion speed was set to 0.5 mm/s [11]. The needle was directed vertically towards the specimen until the cartilage surface was penetrated. Needle insertion was stopped when the load reached 7 N [13]. Apparent articular cartilage thickness was calculated as the difference between the vertical position corresponding to a first variation in the measured compressive load—occurring when the needle tip touched the cartilage surface—and the position consistent with an increase in the slope of force–displacement curve, occurring when the needle tip touched the cartilage/subchondral bone interface [13]. The articular cartilage thickness was calculated by multiplying the apparent articular cartilage thickness by the cosine of the surface angle (see Appendix B) between the vertical direction and the direction normal to the surface [13]. Cartilage thickness in the geometric center was estimated as the mean of the thickness values measured in the four corners (estimated thickness). Then, the articular cartilage thickness was measured directly in the geometric center following the above-described procedure (effective thickness). The percentage prediction error committed by the proposed a priori approach was calculated as the difference between the estimated and effective thickness divided by the effective thickness, expressed as a percentage. One hundred thickness evaluations were performed.

### 2.3. Optimization of Grid Spacing

When performing automated mapping of articular cartilage for its mechanical characterization, it is crucial to define the proper minimum distance on a square grid, or grid spacing, avoiding mutual influence between nearby indentations. As a rule, the grid spacing is set equal to a multiple of the radius (*a*) of the contact area between the articular cartilage and the spherical indenter. Thus, an experimental assessment was employed to investigate whether an indentation (perturbing indentation) performed at three different distances (*a*, 2*a*, 3*a*) away from the indentation point might affect the response of the cartilage in the measuring indentation point (measuring indentation).

The indenter diameter and indentation depth determine the value of *a*. A spherical indenter of 8 mm in diameter was selected for this series. This value is greater than those used in the published reports; in fact, it represents the worst-case scenario as it determines the maximum extension of articular cartilage compression. In order to maximize the effect of the perturbing indentation, the indentation depth was set to 0.25 mm, 25% greater than what is commonly used in the literature (0.20 mm) [7,13,14,29]. Considering a sphere-plane Hertzian contact [30], the value of the contact radius *a* was equal to 1.4 mm. Perturbing indentations were performed at 1.4 mm (*a*), 2.8 mm (2*a*), or 4.2 mm (3*a*) away from the indentation point. The distance between two adjacent rows or columns was 6 mm—i.e., 4*a* rounded to the upper integer—to avoid any interference among indentation points. The resulting indentation scheme is shown in Figure 1.

The described approach entails four indentations on each measuring indentation point, one without (noPI) and three with a preliminary perturbing indentation performed at a distance *a* (PI*a*), 2*a* (PI2*a*), or 3*a* (PI3*a*). Indentation series on measuring indentation points were repeated after a resting period of 40 min [23]. In order to minimize the bias due to change in cartilage response over repeated indentations, a preliminary indentation was performed to precondition the tissue in each measuring indentation point. Additionally, a full counterbalanced order was adopted in selecting the distance from the indentation point where to perform the perturbing indentation. Accordingly, the total number of permutations was 24 (4!, i.e., the factorial of four). A total of 72 points (measuring indentations) were evaluated to achieve three times the number of total permutations and guarantee the full counterbalanced order. All indentations were performed in displacement control in the direction normal to the surface (see Appendix B). The indentation rate was set to 0.25 mm/s. Then, the indenter was held in position for 120 s before releasing from the articular cartilage. When the perturbing indentation was planned, measuring indentation was performed 60 s after the perturbing one.

The first part of the load–displacement curve, up to the load peak, was evaluated by applying the Hayes model to calculate the instantaneous elastic modulus (E_0_) [31]. Poisson’s ratio was set to 0.45 [32]. The load–time curve (load relaxation) was fitted by a stretched exponential function [33,34,35], and the three characterizing parameters—S_0_, τ, and β, where S_0_ is the maximum load value reached prior to load relaxation, τ is the time constant, and β is the stretching parameter—were calculated. E_0_, S_0_, τ, and β values, achieved with a preliminary perturbing indentation, were normalized with the corresponding values achieved in the same measuring indentation point without performing a perturbing indentation (noPI).

### 2.4. Effect of Indenter Diameter on Articular Cartilage Response

In order to cover the whole range used in previous studies evaluating articular cartilage mechanics, seven spherical indenters, with diameter spanning from 1 to 6 mm, with 1 mm increments, and an additional indenter of 8 mm were selected in this study (Appendix A) [7,10,11,12,13,14,15]. Grid spacing was set to 6 mm (see Section 2.3). An a priori estimate of the articular cartilage thickness at the indentation point was performed (see Section 2.2). Aiming to produce the same extent of deformation regardless of the articular cartilage thickness, a nominal deformation corresponding to 15% of the tissue thickness was imposed [36,37]. The deformation rate was set to 0.15 s^−1^ [38,39,40].

Seven indentations, with indenters differing in size, were performed on each indentation point. As in the previous test, a preliminary indentation was performed to precondition the tissue. Then, the seven indentations were performed at 40 min time intervals. Since the number of possible permutations was 5040 (7!, i.e., the factorial of seven), it was not possible to adopt a full counterbalanced order. Therefore, the fully counterbalanced order adopted here was limited to the first two indentations of each series of seven indentations, achieving a number of permutations limited to 42. For the remaining five indentations of each series, the indentation order was defined to ensure that each indenter was placed the same number of times, i.e., six, in each of the remaining last five positions. A total of 126 points were evaluated to investigate three times 42 respecting the above-described scheme, and thus to minimize the bias due to change in cartilage response over repeated indentations.

The load–displacement curve up to the load peak was evaluated by applying the Hayes model, where Poisson’s ratio was set to 0.45, to calculate the instantaneous elastic modulus (E_0_).

### 2.5. Pilot Study on a Human Tibial Plateau

A pilot trial was performed on a human tibial plateau (donor gender: male; donor age: 50 years old) to assess the comparability and repeatability of the indentation experimental procedure. The tibial plateau was processed by following the same procedure reported for bovine knee samples management, and constrained to the X-Y motorized table of the testing machine (Mach-1 V500css, Biomomentum Inc., Laval, QC, Canada, see Section 2.1 and Appendix A).

Indentation parameters were defined based on the results of the tests performed on bovine articular cartilage. Therefore, 6 mm and 8 mm indenters were used to map the articular cartilage with a grid spacing equal to 3*a*, calculated for the 8 mm indenter. The orthogonal grid was overlaid to the tibial plateaus, identifying a total of 15 points on the articular cartilage surface. An a priori estimate of the articular cartilage thickness at the indentation points was performed. All indentations were performed in displacement control to a depth of 0.20 mm or 15% of the estimated articular cartilage thickness, whichever was greater, in the direction normal to the surface (see Appendix B). Two deformation rates were used: 0.15 s^−1^ and 0. 30 s^−1^.

The described approach entails four indentations on each indentation point. As previously reported, a preliminary indentation was performed to precondition the tissue. Then, the indentation series were repeated after a resting period of 40 min. The indenter diameter and the deformation rate were selected randomly for each indentation point.

The load–displacement curve, limited to the indentation depth of 0.20 mm, was evaluated by applying the Hayes model, with Poisson’s ratio set to 0.45, to calculate the instantaneous elastic modulus (E_0 0.20mm_). The same evaluation to calculate the instantaneous elastic modulus (E_0 15%def_) was performed on the load–displacement curve limited to a nominal deformation of 15%, determined by using the a priori estimate of the articular cartilage thickness.

### 2.6. Data Analysis

Fitting of experimental data was performed choosing a specific mathematical model to describe the data trend—i.e., the Hayes’ model or the stretched exponential function (see Appendix C)—and determining function parameters by minimizing the root mean square error by using a numerical computing environment (MATLAB R2021a, MathWorks, Natick, MA, USA).

Paired cartilage thickness data, i.e., the estimated and effective thickness value for each investigated point, were analyzed by using the paired-sample Wilcoxon test performed using a commercial software (SPSS version 14.0.1, SPSS Inc., Chicago, IL, USA).

E_0_, S_0_, τ, and β normalized values achieved after perturbing indentations were analyzed by using the Friedman test, followed by the Nemenyi post hoc test, both performed using a free software environment (R version 4.1.2, R Core Team, R: A Language and Environment for Statistical Computing, Vienna, Austria, https://www.R-project.org/). This statistical analysis was also used to identify differences among E_0_ values determined by indenting the articular cartilage with indenters of different size.

The paired-sample Wilcoxon test was also used to identify differences between E_0_ values determined by indenting the articular cartilage with indenters of different size, two indentation rates, and two applied deformations on the human tibial plateau in the pilot study.

## 3. Results

### 3.1. A Priori Estimate of the Articular Cartilage Thickness at the Indentation Point

We investigated 100 different points on the articular cartilage surface for a total of 500 needle indentations (400 indentations performed in the corners and 100 in the geometric centers of 100 squares). The achieved cartilage thickness values measured in the geometric centers fell in the range of 0.5–2.4 mm. The distributions of the estimated and effective thickness values are shown in Figure 2.

No statistical difference was found between two distributions (paired-sample Wilcoxon test, *p* = 0.56). The 5th, 25th, 50th, 75th, and 95th percentile of the percentage prediction error distribution was −10.6%, −4.7%, 0.5%, 6.5%, and 16.4%, respectively. The complete dataset is provided as Appendix A.

### 3.2. Optimization of Grid Spacing

Four indentations were performed on each of the 72 points for a total of 288 indentations. The execution of a preliminary indentation (perturbing indentation), close to the indentation point where tissue properties were going to be investigated, changed the response of the articular cartilage. Indeed, the Friedman test showed a significant difference among the four groups (noPI, PI*a*, PI2*a*, PI3*a*) regardless of the analyzed parameter (*p* < 0.001 for all four parameters E_0_, S_0_, τ, and β). The variation in tissue response depended on the distance from the preliminary perturbing indentation. The Nemenyi post hoc test showed a significant difference between noPI and PI*a* regardless of the analyzed parameter (*p* < 0.001 for all four parameters E_0_, S_0_, τ, and β). Conversely, the post hoc test showed a significant difference between noPI and PI2*a* only for S_0_ (*p* < 0.001). A significant difference was also found between noPI and PI3*a* for τ (*p* < 0.001) and β (*p* = 0.01). The distributions of normalized values of E_0_, S_0_, τ, and β for Pi*a*, PI2*a*, and PI3*a* groups are shown in Figure 3. The effect of the perturbing indentation was also marked by the increase in data dispersion associated with the decreasing distance from the preliminary perturbing indentation. The complete dataset is provided as supplementary material (Appendix A).

### 3.3. Effect of Indenter Diameter on Articular Cartilage Response

Seven indentations were performed on each of the 126 points for a total of 882 indentations. The distributions of E_0_ values determined with indenters having different diameter are shown in Figure 4. The indenter diameter affected the instantaneous response of the articular cartilage: the Friedman test showed a significant difference among the seven groups (*p* < 0.001). Collected data showed an increase in E_0_ by increasing the indenter diameter up to 6 mm (Nemenyi post hoc test: *p* < 0.001 for all comparison between E_0_ values determined with 6 mm indenter and E_0_ values determined with indenter smaller than 6 mm). The same results were achieved comparing E_0_ values determined with a 8 mm indenter and E_0_ values determined with an indenter smaller than 6 mm. Conversely, no significant difference was found comparing E_0_ values determined with 6 mm and 8 mm indenters. The complete dataset is provided as Appendix A.

### 3.4. Pilot Study on a Human Tibial Plateau

The estimated cartilage thickness of the 15 points identified by overlaying the orthogonal grid to the tibial plateau fell in the range 0.9–2.9 mm with a median of 1.5 mm. The 5th, 25th, 50th, 75th, and 95th percentile of the distribution of the indentation depth imposed to achieve a 15% nominal deformation was 0.15 mm, 0.020 mm, 0.23 mm, 0.29 mm, and 0.44 mm, respectively.

The distributions of E_0_ values determined with different testing conditions are shown in Figure 5.

No significant differences were found between E_0_ values determined with 6 mm and 8 mm indenters. Conversely, the paired-sample Wilcoxon test showed a significant increase in E_0_ values by doubling the indentation rate regardless of the testing conditions (*p* < 0.001). Moreover, a significant difference was found between the distribution of E_0 0.20mm_ and E_0 15%def_ values, achieved with the same testing conditions (*p* = 0.03, Appendix A). Additionally, the dispersion of E_0 0.20mm_ values was in general 5–6% greater than E_0_ 15%def values. The complete dataset is provided (Appendix A), and changes in E_0_ in relation to the investigated testing parameters are highlighted (Appendix A).

## 4. Discussion

We aimed to optimize the indentation procedure for mapping the mechanical properties of articular cartilage when supported by underlying bone tissue, i.e., the response of the whole osteochondral unit to indentation with a spherical indenter. The main novelty of the proposed approach was to perform the indentation tests by controlling the nominal deformation and, consequently, the deformation rate imposed to the articular cartilage while optimizing two indentation parameters such as grid spacing and indenter size.

The proposed approach requires the a priori knowledge of the cartilage thickness. Different methods have been proposed to measure it, the most accurate one entailing needle probing [13,28,41,42,43,44,45,46]. This method has been carried out after the indentation test since the indentation protocol commonly relies on the application of a constant displacement, or indentation depth, regardless of the thickness of the articular cartilage [7,11,12,13,22,23]. Additionally, the direct measurement cannot be performed before the indentation test because it creates a discontinuity in the cartilage tissue at the indentation point. However, the articular cartilage thickness can be estimated by measuring the thickness in nearby positions, as shown in this study. The achievable accuracy (see Section 3.1) is comparable to, or even better than, that reported for non-contact methods (stereomicroscopy, range 7.9–10.5% [42]; ultrasonography, range 0–31% [45,47]). Therefore, the estimation of articular cartilage thickness, on the basis of thickness measurements carried out in the surrounding area of the indentation point, could be an alternative to non-contact methods.

Mapping the mechanical properties of articular cartilage surfaces requires the definition of a grid to be superimposed to the joint surface. The definition of the optimal grid spacing would allow us to achieve the maximum resolution of the gridded cartilage surface properties, minimizing any mutual influence between nearby indentations. In the literature, some indentation-based studies mapping articular cartilage properties reported the number of the positions per unit surface area—or the density—of the superimposed grid, without any consideration of the possible mutual influence between nearby indentations [13,46]. However, few studies investigating cartilage mechanics at the nano- or micro-scale reported that the minimum distance between subsequent points should be at least three times (3*a*) the contact radius resulting between the probe and the cartilage surface [19,21]. Consistently, our findings show that decreasing the grid spacing to 2*a* increases the data dispersion and statistically alters the cartilage instantaneous response. A reduction in the grid spacing to a further (*a*) increases data dispersion and, more importantly, significantly alters the whole mechanical response of the articular cartilage. The observed effect could be explained considering that indentation determines interstitial fluid flow through the tissue in the surrounding area. Therefore, any further indentation performed too close to the previously indented point, and before tissue equilibrium has been recovered, might be affected by locally increased fluid content, as suggested by the trend of τ and β found decreasing the distance of the preliminary perturbing indentation from the indentation point. Looking at the collected values, it could be argued that the 2*a* grid spacing represents an acceptable compromise between accuracy and grid resolution. However, the root mean square percentage error (RMSPE) values—calculated comparing the 72 stretched exponential functions describing the response of the cartilage without perturbation (noPI) with each of the stretched exponential functions describing the response of the perturbed cartilage (PI*a*, PI2*a*, PI3*a*) in the same 72 measuring points—progressively decrease by increasing the distance between the two points where the perturbing and measuring indentation were performed; accordingly, a minimum extent of RMSPE is reached considering a 3*a* grid spacing for all the investigated points (Appendix A). Therefore, it seems recommendable to set the grid spacing to 3*a* to minimize the bias induced by mutual influence between nearby indentations in the articular cartilage response.

As for the choice of the indenter diameter, at a first glance, the wide range of E_0_ values is noteworthy. Such a wide range was somehow expected since tissue samples were collected from both femoral condyles and tibial plateaus of ten bovine knees. Despite the wide range of E_0_ values, the present results highlighted an increasing trend in E_0_ values as a function of indenter diameter up to 6 mm. Neglecting the studies carried out at nano- and micro-scale [14,48], since indentation length scale affects the response of the articular cartilage [49], such an effect was partially investigated in two studies [18,19]. Although in the cited studies, the measured parameters were different or calculated using a different approach, a partial comparison is still possible. Chandwadkar found an apparent increasing trend in peak load values increasing the indenter diameter from 1 mm to 5 mm by 2 mm at a time [19]. Moore et al. found that doubling the indenter diameter (from 3.175 mm to 6.350 mm) led to a rise in the cartilage contact modulus, which includes contributions from fluid and solid stresses [18]. Therefore, both studies do not contradict our findings. Moore and colleagues suggested that increasing the indenter radius, while maintaining the indentation depth fixed (indentation depth: 0.15 mm), increases the interaction depth in the assumed Hertzian contact. The articular cartilage is not homogeneous, showing higher water and lower glycosaminoglycan content in the surface layer. Therefore, decreasing the indenter diameter would make the tissue response more surface-sensitive. A further contribution might be ascribed to the interstitial fluid: the greater the indenter diameter, the greater the tissue volume, with its fluid content, perturbed by indentation. This might change the extent of the contribution of the extracellular matrix and interstitial fluid in supporting the external load. These considerations do not necessarily imply that small indenters must be avoided. However, the reported bias in the instantaneous response of articular cartilage must be considered when comparisons with other studies are made, taking the indenter diameter into account.

Evaluating the human articular cartilage response through indentation test and, above all, aiming at defining a standard indentation protocol, which would allow the comparison between different studies, requires us to properly identify the main peculiarities of the indentation procedure. A nominal deformation corresponding to 15% of the tissue thickness instead of a fixed indentation depth was applied to induce the same nominal deformation regardless of the cartilage thickness and considering the contact features between highly congruent surfaces and the average condition peculiar to the human knee joint [36,37]. E_0_ values determined by indenting human articular cartilage fell within the range reported in the literature [7,13,35,50]. The pilot study carried out on a human tibial plateau showed no differences between E_0_ values achieved by indenting articular cartilage with 6 mm and 8 mm diameter indenters, regardless of the rate or applied indentation, in agreement with the data collected by testing bovine cartilage. The experimental procedure was sensitive to the deformation rate. It is well known how such a parameter—also known as indentation rate, loading rate or displacement rate—impacts the tissue response [8,51,52]. Moreover, the dependence of the cartilage behavior on the deformation rate was detected at the millimeter [18], micro- [14,53], and nano-scale [53,54]. From this perspective, and independent from the investigated scale, the rate at which the tissue is indented directly impacts the cartilage response. Accordingly, our results highlighted a statistical increase in the cartilage elastic response, i.e., E_0_, by raising the deformation rate from 0.15 s^−1^ to 0.30 s^−1^, regardless of the applied indentation (15% of the thickness or 0.2 mm). However, the amount of deformation applied to the articular cartilage impacts the articular cartilage response. A significant difference was found between E_0 0.20mm_ and E_0 15%def_ values, the values of the latter being slightly higher. It is worth noting that the median value of the distribution of the indentation depth imposed to achieve a 15% nominal deformation was 0.23 mm, i.e., slightly higher than the fixed value for the indentation depth of the E_0 0.20mm_ group [12,55,56,57]. Additionally, maintaining the nominal deformation of the cartilage thickness fixed at 15% seems to reduce dispersion. Reduction in data dispersion would allow us to improve the statistical power in detecting a correlation among different parameters and, more in general, to provide more accurate information for tissue engineering aimed at developing a complex composite scaffold for articular cartilage repair.

Some limitations must be highlighted. First, the pilot study was carried out on one human tibial plateau. A larger sample size would allow us to evaluate more accurately the proposed procedure in terms of data dispersion of parameters describing the cartilage response. Second, the deformation-based protocol here proposed relies on the a priori assessment of cartilage thickness. From this perspective, a procedure based on the evaluation of such a parameter in nearby positions was implemented, specifically by applying the needle technique [41]. This technique creates four discontinuities, i.e., four small holes, around the indentation point of the order of magnitude of half a millimeter. The need for measuring cartilage thickness could be overcome by operating in pressure control mode. This alternative approach might also determine a reduction in data dispersion. Nevertheless, indenting articular surfaces by a pressure-based protocol is not yet feasible due to technological limitations; such an approach would require an accurate sensor capable of measuring the contact pressure at the indenter/cartilage interface on a small area of the indenter surface located on the indenter axis.

## 5. Conclusions

Aiming to fulfill tissue engineering approaches treating pathologies affecting articular cartilage, a sound assessment of the cartilage behavior must be provided. The present study represents a step forward in properly evaluating how grid spacing, indenter diameter, and induced deformation by indentation all impact the articular cartilage response. Grid spacing should be set to 3*a* to minimize bias induced by indentations performed in adjacent grid points. Indenter diameter should be set to least at 6 mm to avert a bias in E_0_ values and to allow data comparison among different studies. When grid spacing and indenter diameter are selected to minimize bias in articular cartilage response, the deformation-based approach—15% of the cartilage thickness—represents a reliable method to investigate articular cartilage behavior by indentation test aiming at reducing data dispersion. The proposed procedure would allow a sound assessment of the articular cartilage mechanical properties to be compared with those of complex composite scaffolds.

## Figures and Tables

**Figure 1 materials-15-06425-f001:**
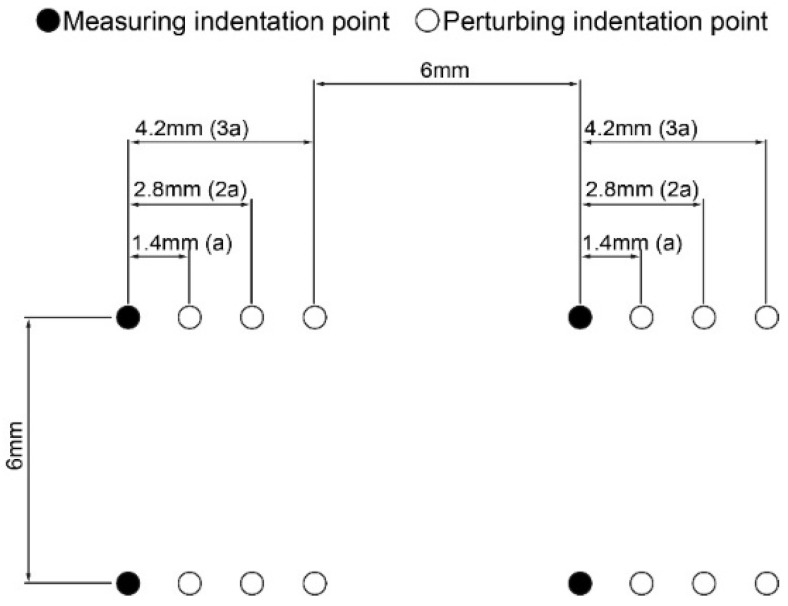
Scheme of the orthogonal grid used to evaluate the effect of perturbing indentations on the response of articular cartilage in the measuring indentation points.

**Figure 2 materials-15-06425-f002:**
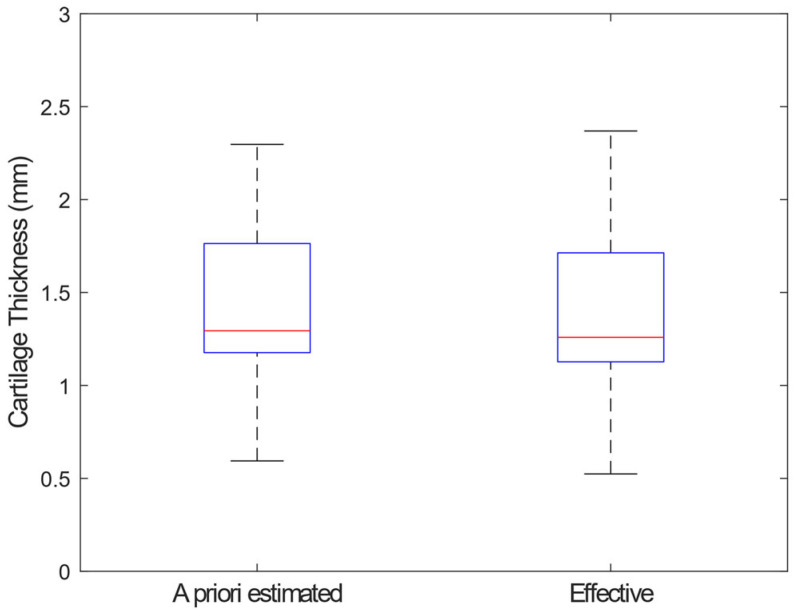
Box plot of the estimated and effective thickness values.

**Figure 3 materials-15-06425-f003:**
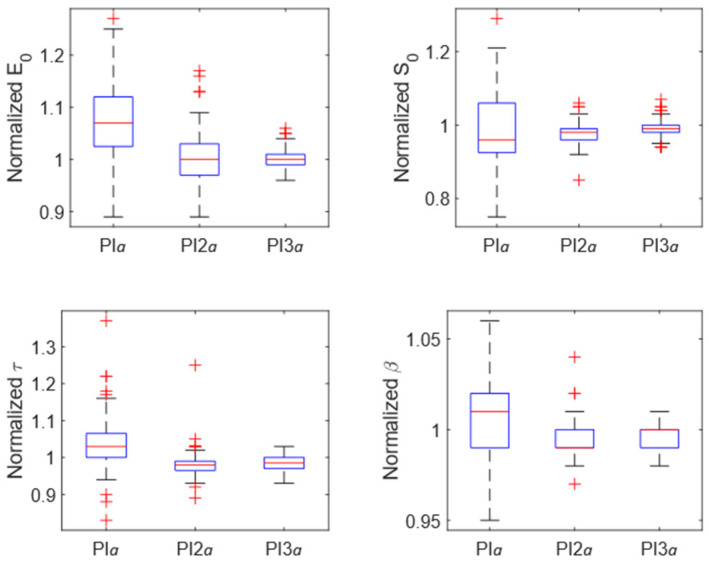
Box plots of the normalized values of E_0_ (**top left**), S_0_ (**top right**), τ (**bottom left**), and β (**bottom right**) for PI*a*, PI2*a*, and PI3*a* groups.

**Figure 4 materials-15-06425-f004:**
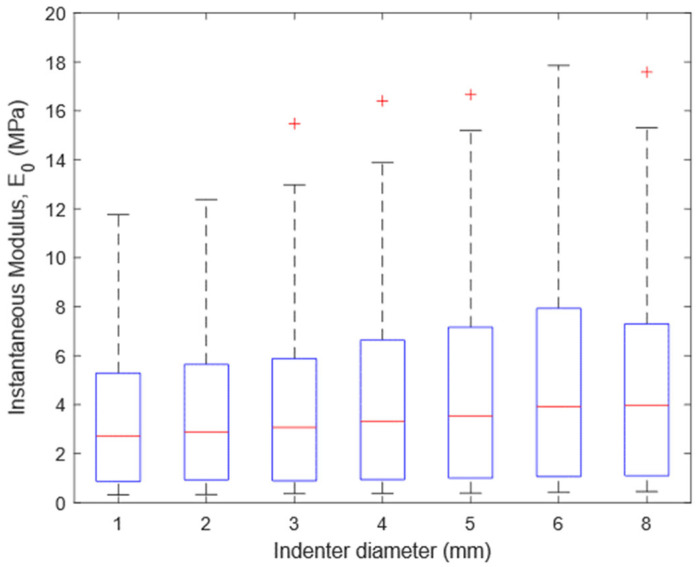
Box plot of E_0_ values determined with indenters having different diameters, achieved by evaluating 126 points over the articular surface of ten bovine knees.

**Figure 5 materials-15-06425-f005:**
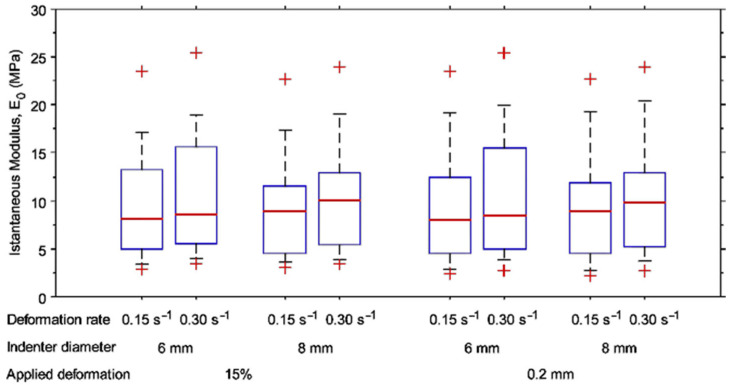
Box plots of E_0 15%def_ and E_0 0.20mm_ values determined using 6 mm and 8 mm indenters and of E_0_ to indent the articular cartilage of the tibial plateau at two different deformation rates (0.15 s^−1^ and 0.30 s^−1^).

## Data Availability

Data supporting reported results are available in the Appendix A.

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
