# Peer review of "Optimization of In Situ Indentation Protocol to Map the Mechanical Properties of Articular Cartilage"

_materials, 2022, doi:10.3390/ma15186425_

Round 1

Reviewer 1 Report

Please create a shorter title as now it looks rather as a summary

The abstract seems ok, but there is difficult to understand why certain condition was selected. Here I propose giving actual link between the physics and your statements

From your introduction it is rather difficult to understand the need of this wok

The criticism used in this part seems rather brief and not very clear

Please avoid block citations “[7, 10, 11, 12, 13, 14, 15].” Also this apply for other one use in this work ..max 3 it will be enough

The scientific novelty is not highlighted in an appropriate manner

Please provide the ethical consent for using animal; the same apply for the human !

The standard deviation in Figure 4 shows very large dispersion – how do you explain it ? cause you said that you conducted 7 indentation for each point, normally I expect the std to be minimal ..

The results are very little interpreted

Your conclusion is just a statement it requires a proper link between the results and conclusion and there in conclusion are required some quantitative details

Author Response

We thank the reviewers for their comments and suggestions. We have tried to address each point. Please find enclosed our replies. The changes are marked in red colour in the revised manuscript.

Reviewer 1

  1. Please create a shorter title as now it looks rather as a summary

The title has been revised accordingly.

  1. The abstract seems ok, but there is difficult to understand why certain condition was selected. Here I propose giving actual link between the physics and your statements

Abstract has been revised by adding the range of the investigated parameters; nevertheless, a complete dissertation on the conditions selected in the study are reported in the Introduction section due to the word limit peculiar of the abstract. 

  1. From your introduction it is rather difficult to understand the need of this wok

The Introduction section has been revised trying to clarify the open issues.

  1. The criticism used in this part seems rather brief and not very clear

The Introduction section has been revised, aiming to better highlight gaps to fulfill regarding articular cartilage indentation.

  1. Please avoid block citations “[7, 10, 11, 12, 13, 14, 15].” Also this apply for other one use in this work ..max 3 it will be enough

We have summarize block citations as suggested.  Please note that in some cases we have left more than 3 references because we think that the mentioned references are necessary to support our statements or main findings of the study.

  1. The scientific novelty is not highlighted in an appropriate manner

Introduction section has been revised aiming to highlight in an appropriate manner the scientific novelty of the study.

  1. Please provide the ethical consent for using animal; the same apply for the human !

The ethical consent for using human tissues has been reported in the manuscript (see Institutional Review Board Statement section). Regarding animal tissues, they were collected from animals killed for alimentary purposes, as stated in the manuscript (see paragraph 2.1). The use of tissues collected from animals killed for alimentary purposes does not require ethical consent.

  1. The standard deviation in Figure 4 shows very large dispersion – how do you explain it ? cause you said that you conducted 7 indentation for each point, normally I expect the std to be minimal ..

Figure 4 concerns the results achieved by indenting 126 points on the articular cartilage surface of multiple samples extracted from both femoral condyles and tibial plateaus of ten bovine knees (see lines 108-113, 225-227, 314-315). Accordingly, the wide variation corresponding to each boxplot, i.e., of each column plot of Figure 4, is related to the variability of the articular cartilage instantaneous response (E0) over the knee articular surface, to date reported only considering human tibio-femoral joint. Such information has been added in the manuscript (see lines 406-408). Caption of Figure 4 has been revised too. 

  1. The results are very little interpreted

Discussion section has been revised accordingly.

  1. Your conclusion is just a statement it requires a proper link between the results and conclusion and there in conclusion are required some quantitative details

Conclusions have been revised accordingly. 

Reviewer 2 Report

The authors performed indentation procedure for mapping the mechanical properties of articular cartilage the research is so interesting however, the manuscript need some modifications to be accepted for publication.

 On Line 39  the authors should review the sentence “accounting for its site-, depth- , and time-dependent behavior”  something is missing.

 The author do not provide any picture about bovine knees used samples

The authors should provide pictures of seven spherical indenters used in the experiment. So they have to explain clearly the load applied in every indentation  as well as the equipment and the standard used.

The authors should explain clearly how was the preparation of the specimen for indentation in human tibial plateau  as well as provide picture and load applied.

 Due to authors are Mapping the mechanical properties of articular cartilage, they have to show the results in a picture to evidence how change the mechanical properties in every point of the cartilage

Author Response

We thank the reviewers for their comments and suggestions. We have tried to address each point. Please find enclosed our replies. The changes are marked in red colour in the revised manuscript.

Reviewer 2

The authors performed indentation procedure for mapping the mechanical properties of articular cartilage the research is so interesting however, the manuscript need some modifications to be accepted for publication.

  1. On Line 39 the authors should review the sentence “accounting for its site-, depth- , and time-dependent behavior” something is missing.

The missing term has been added.

  1. The author do not provide any picture about bovine knees used samples

Picture of a bovine knee, along with the one of a tested sample, has been provided as supplementary materials (see Supplementary Figure 1).

  1. The authors should provide pictures of seven spherical indenters used in the experiment. So they have to explain clearly the load applied in every indentation as well as the equipment and the standard used.

Picture of the seven spherical indenters used in the experiments has been provided as supplementary material (see Supplementary Figure 2). Regarding how indentation test was conducted, and aiming at producing the same extent of deformation regardless of the articular cartilage thickness, a displacement control was applied after estimating local articular cartilage thickness. Accordingly, a nominal deformation corresponding to 15% of the tissue thickness was imposed (see lines 212-214). Regarding the equipment used in the experiments, details are reported in paragraph 2.1 and 2.5. Concerning articular cartilage mechanical assessment, the only available standard was ASTM F2451-05, withdrawn on 2019. Accordingly, and considering the relative literature, we proposed an experimental method capable of indenting articular cartilage by inducing a nominal deformation close to the one peculiar of the in vivo condition.

  1. The authors should explain clearly how was the preparation of the specimen for indentation in human tibial plateau as well as provide picture and load applied.

The requested details have been added, as well as an additional picture of the experimental setup with the tibial plateau (see Supplementary Figure 3). Please note that the test was conducted in displacement control, thus to apply a constant tissue deformation as described in the text.

  1. Due to authors are Mapping the mechanical properties of articular cartilage, they have to show the results in a picture to evidence how change the mechanical properties in every point of the cartilage

The extent of the investigated property, together with their variation achieved by changing specific protocol parameters – i.e., indenter diameter, and deformation rate – was reported for each point evaluated as supplementary materials (see Supplementary Table S4). Moreover, bar graphs (Supplementary Figure S5 to S16) have been added as supplementary materials aiming to highlight better the change of the articular cartilage instantaneous modulus in relation to the investigated testing parameters, i.e., indenter diameter, indentation rate, and applied deformation.

Reviewer 3 Report

The manuscript should be accepted after following minor changes.

        i.            The novelty of the paper should be explained in more detail (lines 85-92).

      ii.            In section 2.6, the mathematical model and its solution procedure should be explained in more detail.

    iii.            How the results are validated? A comparison table/figure should be added. 

Author Response

We thank the reviewers for their comments and suggestions. We have tried to address each point. Please find enclosed our replies. The changes are marked in red colour in the revised manuscript.

Reviewer 3

The manuscript should be accepted after following minor changes.

  1. The novelty of the paper should be explained in more detail (lines 85-92).

The Introduction section has been revised extensively aiming to evidence better the novelty of the paper.

  1. In section 2.6, the mathematical model and its solution procedure should be explained in more detail.

The mathematical models have been added in Appendix B, to let the revised text flow smoothly. Please note that details about solution procedures are already reported at the end of each paragraph where they have been used thus to make the text easier for the reader (see paragraphs 2.3, 2.4, and 2.5).

  1. How the results are validated? A comparison table/figure should be added.

Validation is not possible since cartilage samples with known properties are not available. However, an indirect comparative validation is possible by comparing the present findings with those reported in previous studies. Focusing on validating the results achieved by indenting bovine knee articular cartilage, to date no studies investigated its instantaneous modulus. Regarding how indenter diameter impact on the articular cartilage elasticity, literature do not contradict our findings, highlighting an increase of its extent by increasing the indenter diameter (see references 18 and 19). Concerning indentation on human tibial plateau, first, the range of E0 values is supported by relative literature (see references 7, 13, 35, 50).  Second, such a test allowed to validate findings achieved by testing bovine knee articular cartilage, i.e., no significant differences between E0 values determined with 6mm and 8mm indenters. Finally, the dependance of the articular cartilage behavior on the deformation rate highlighted by our results is supported by previous studies (see references 8, 14, 50-53). The comparative validation has been reported in the Discussion section.

Round 2

Reviewer 2 Report

no one